# Melatonin Use during Pregnancy and Lactation Complicated by Oxidative Stress: Focus on Offspring’s Cardiovascular–Kidney–Metabolic Health in Animal Models

**DOI:** 10.3390/antiox13020226

**Published:** 2024-02-12

**Authors:** You-Lin Tain, Chien-Ning Hsu

**Affiliations:** 1Division of Pediatric Nephrology, Kaohsiung Chang Gung Memorial Hospital, Kaohsiung 833, Taiwan; tainyl@cgmh.org.tw; 2Institute for Translational Research in Biomedicine, Kaohsiung Chang Gung Memorial Hospital, Kaohsiung 833, Taiwan; 3College of Medicine, Chang Gung University, Taoyuan 333, Taiwan; 4Department of Pharmacy, Kaohsiung Chang Gung Memorial Hospital, Kaohsiung 833, Taiwan; 5School of Pharmacy, Kaohsiung Medical University, Kaohsiung 807, Taiwan

**Keywords:** oxidative stress, hypertension, metabolic syndrome, developmental origins of health and disease (DOHaD), kidney disease, nitric oxide, melatonin, antioxidants

## Abstract

Cardiovascular–kidney–metabolic (CKM) syndrome has emerged as a major global public health concern, posing a substantial threat to human health. Early-life exposure to oxidative stress may heighten vulnerability to the developmental programming of adult diseases, encompassing various aspects of CKM syndrome. Conversely, the initiation of adverse programming processes can potentially be thwarted through early-life antioxidant interventions. Melatonin, originally recognized for its antioxidant properties, is an endogenous hormone with diverse biological functions. While melatonin has demonstrated benefits in addressing disorders linked to oxidative stress, there has been comparatively less focus on investigating its reprogramming effects on CKM syndrome. This review consolidates the current knowledge on the role of oxidative stress during pregnancy and lactation in inducing CKM traits in offspring, emphasizing the underlying mechanisms. The multifaceted role of melatonin in regulating oxidative stress, mediating fetal programming, and preventing adverse outcomes in offspring positions it as a promising reprogramming strategy. Currently, there is a lack of sufficient information in humans, and the available evidence primarily originates from animal studies. This opens up new avenues for novel preventive intervention in CKM syndrome.

## 1. Introduction

Increasing recognition of the pathological interconnections among metabolic risk factors like obesity and diabetes, chronic kidney disease (CKD), and cardiovascular disease (CVD) has given rise to the conceptualization of cardiovascular–kidney–metabolic (CKM) syndrome [1]. For the first time, the 2023 Scientific Statement from the American Heart Association defines CKM syndrome as a systemic disorder marked by pathophysiological interactions among metabolic risk factors, CKD, and the cardiovascular system. This interaction results in multiorgan dysfunction and a heightened risk of adverse cardiovascular and renal outcomes [1]. CKM syndrome is classified into four distinct stages, ranging from stage 0 to stage 4. These stages likely represent varying degrees of progression and severity within the complex spectrum of this disease. Various key components manifest at different stages, contributing to the varied degrees of progression and severity within the complex spectrum of CKM syndrome.

While the global disease burden of individual metabolic diseases, CKD, and CVD is widely recognized [2,3,4,5], the complex and mutually reinforcing detrimental relationships among them remain largely unclear. Epidemiological studies suggest a multidirectional connection among these conditions. For instance, individuals with CKD face an elevated risk of developing CVD [6]. Comorbidities associated with CKD, such as obesity, diabetes, and hypertension, constitute significant components of metabolic syndrome [7]. Additionally, cardiorenal syndrome is a recognized condition in which dysfunction of either the heart or kidneys can impair function in the other organs [8]. Furthermore, metabolic syndrome, a cluster of conditions, collectively amplifies the susceptibility to CVD [9]. Recently, the concept of metabolic syndrome has evolved into one of cardiometabolic syndrome, encompassing characteristics of metabolic syndrome along with traditional CVD factors [10]. Consequently, it is plausible that CVD, CKD, and metabolic diseases may interact at the pathophysiological level, leading to clinical overlap between these health conditions [11].

Approximately 40% of adults in the United States are estimated to be affected by CKM syndrome [12]. Given that CKM syndrome impacts nearly all organ systems, there is a substantial global burden of compromised cardiovascular–kidney–metabolic health. Although managing the entire syndrome holistically, rather than focusing on individual diseases, is recommended for optimal care [12], a comprehensive therapeutic guideline is still pending. It is noteworthy that placing emphasis on early prevention has the potential to alleviate the burden associated with CKM syndrome. Recognizing the interplay between CKM diseases is crucial for adopting a more holistic approach to CKM care, moving beyond the isolated treatment of individual conditions. This broader perspective holds promise for enhancing global health outcomes in the future.

An expanding body of literature indicates that unfavorable environmental conditions during pregnancy and early infancy may increase the susceptibility to adult diseases [13], encompassing all facets of CKM syndrome. Termed the Developmental Origins of Health and Disease (DOHaD), this theory posits that a developing fetus responds to environmental challenges by making structural and functional adaptations, thereby increasing the risk of developing chronic diseases later in life [14]. Although our understanding of the pathophysiological mechanisms of DOHaD remains incomplete, oxidative stress plays a role in the etiology of many DOHaD-related diseases [15].

Oxidative stress arises from an imbalance between the production of harmful reactive oxygen/nitrogen species (ROS/RNS) and the antioxidant system’s ability to detoxify them [16]. Both human and experimental studies provide evidence supporting the involvement of oxidative stress in various aspects of CKM syndrome [17,18,19] (Figure 1). Conversely, the administration of antioxidants has been reported to ameliorate CKM syndrome [20,21,22]. Early-life interventions, referred to as reprogramming, have the potential to reverse detrimental programming processes and alleviate the onset of adult diseases [23]. The use of antioxidants during pregnancy and lactation has demonstrated benefits as a reprogramming strategy for preventing numerous adult diseases [24].

Melatonin, primarily secreted by the pineal gland during nighttime, serves as a versatile hormone [25]. Alongside its metabolites, melatonin exhibits antioxidant properties, positioning this molecule as an intrinsic safeguard against various disorders associated with oxidative stress [26,27]. In recent times, melatonin has found applications in the treatment of pregnant women and various pediatric conditions [28,29,30]. In this context, we will delve into the existing knowledge regarding the links among melatonin, maternal oxidative stress, and the development of offspring CKM syndrome.

A comprehensive literature review was conducted by querying the databases Embase, MEDLINE, and the Cochrane Library to identify studies published in English. The search utilized relevant keywords pertaining to melatonin, oxidative stress, DOHaD, and CKM syndrome. The employed search terms included “metabolic syndrome”, “hypertension”, “dyslipidemia”, “hyperlipidemia”, “obesity”, “diabetes”, “insulin resistance”, “hyperglycemia”, “liver steatosis”, “chronic kidney disease”, “cardiovascular disease”, “atherosclerosis”, “heart failure”, “cardiorenal syndrome”, “developmental programming”, “DOHaD”, “ free radicals”, “offspring”, “progeny”, “mother”, “prenatal”, “nitric oxide”, “oxidative stress”, “pregnancy”, “reprogramming”, “reactive oxygen species”, “reactive nitrogen species”, and “melatonin”. Additionally, we scrutinized the reference lists of articles to identify any additional references relevant to this review. The search concluded on 30 November 2023.

## 2. Oxidative Stress

### 2.1. ROS, RNS, NO, and Antioxidants

Oxidative stress results from an imbalance between prooxidants and antioxidants [16]. The detrimental effects of oxygen/nitrogen free radicals and non-radical reactive species, collectively referred to as ROS/RNS, can lead to potential biological oxidative damage [31]. Cellular nitric oxide (NO) can interact with ROS, giving rise to various RNS implicated in nitrosative damage. Among the crucial reactive species are the superoxide anion radical (O_2_^−^), hydroxyl radical (OH), hydrogen peroxide (H_2_O_2_), and peroxynitrite (ONOO^−^). Major enzymatic sources of ROS include the electron transport chain complexes in mitochondria, NADPH oxidase (NOX), and xanthine oxidase (XO). Additionally, NO can be produced through three isoforms of NO synthase (NOS): neuronal NOS (nNOS), inducible NOS (iNOS), and endothelial NOS (eNOS) [32]. Asymmetric and symmetric dimethylarginine (ADMA and SDMA, respectively), inhibitors of NOS, have the potential to uncouple NOSs, leading to peroxynitrite generation. This not only diminishes NO production, but also intensifies oxidative stress [33].

Conversely, excessive ROS/RNS can be counteracted by antioxidant systems, encompassing enzymatic components such as superoxide dismutase (SOD) and non-enzymatic antioxidants like glutathione [34]. Throughout pregnancy, a delicate balance is established among ROS, RNS, and antioxidants, crucial for determining normal fetal growth and development.

### 2.2. Oxidative Stress during Pregnancy

Throughout pregnancy, the physiological generation of ROS plays a crucial role in various developmental processes, encompassing oocyte maturation [35], embryo implantation [36], placental differentiation [37], and fetal development [38]. The fetal oxygen requirement varies across different trimesters, with low levels during the initial trimester and a subsequent increase in response to rapid fetal weight gain and the formation of fetal–placental circulation [39].

Compromised pregnancies, marked by maternal illnesses and adverse conditions, are associated with oxidative stress. These conditions include preeclampsia, maternal smoking, obesity, diabetes, placental insufficiency, preterm labor, and intrauterine growth retardation (IUGR) [40]. For instance, current evidence suggests that the pathogenesis of preeclampsia involves increased lipid peroxidation products, elevated nitrotyrosine immunostaining, and reduced antioxidant enzyme activities [41]. Another significant indicator is ADMA. There is growing support for the association of high ADMA levels with conditions such as preeclampsia [42], gestational diabetes [43], IUGR, and placental insufficiency [44]. Furthermore, studies have reported elevated levels of ROS or oxidative stress by-products and decreased antioxidant levels in cases of preterm labor [45].

### 2.3. Animal Models of Oxidative-Stress-Associated Offspring CKM Syndrome

Accumulating evidence underscores the pathogenic interplay between oxidative stress and various facets of CKM syndrome, including obesity [46], kidney disease [47], CVD [48], diabetes [49], nonalcoholic fatty liver disease (NAFLD) [50], and hypertension [51]. However, there is a lack of information regarding oxidative stress during human pregnancy that may be associated with offspring CKM syndrome in childhood or adulthood.

Several recent animal studies emphasize the significance of early-life oxidative stress concerning specific components of CKM syndrome, such as kidney disease or hypertension, and these have been comprehensively reviewed elsewhere [24,52]. However, a notable gap exists in studies directly addressing the impact of early-life oxidative stress on all CKM traits in adult offspring. Therefore, Table 1 predominantly compiles findings from animal models exhibiting at least two components of CKM syndrome in offspring that are associated with oxidative stress [53,54,55,56,57,58,59,60,61,62,63,64,65,66,67,68,69,70,71,72,73,74,75,76,77,78,79,80,81,82,83,84,85,86,87,88,89,90,91,92,93,94,95,96,97,98,99,100]. Moreover, this review focuses on rat models to facilitate relevant comparisons of major CKM syndrome components throughout the lifespan.

Considering that one rat month is roughly equivalent to three human years [101], Table 1 provides the age of offspring outcomes, ranging from 12 to 52 weeks, in rats. This timeline corresponds to humans from young adulthood to middle age.

Imbalances in maternal nutrition play a significant role in inducing oxidative stress-related CKM syndrome in adult offspring. Various nutritional insults contribute to this, including calorie restriction [53,54,55], protein restriction [56,57,58], maternal high-fructose diet [59,60,61,62,63], and maternal high-fat diet [64,65,66,67,68]. Additionally, maternal illnesses and pregnancy complications can disrupt the oxidative balance, leading to the manifestation of CKM syndrome in offspring. For instance, maternal diabetes has been identified as a contributor to oxidative stress, linked to obesity, insulin resistance, hypertension, dyslipidemia, and kidney disease in adult offspring [69,70,71].

Several early-life risks have been associated with the development of CKM syndrome in offspring, including uteroplacental insufficiency [72,73,74,75], maternal chronodisruption [76,77,78], maternal stress [79,80,81,82], maternal chronic kidney disease (CKD) [83,84], nicotine exposure [85,86,87,88], and ethanol exposure [89,90,91]. Furthermore, offspring CKM syndrome can be programmed by dams exposed to environmental chemicals such as 2,3,7,8-tetrachlorodibenzo-p-dioxin (TCDD) [92,93], bisphenol A (BPA) [94,95,96], and di-n-butyl phthalate (DEHP) [99,100,101].

### 2.4. Mechanisms Underlying Oxidative Stress in CKM Syndrome

In developmental-origin-related CKM syndrome, a comprehensive array of oxidative stress mechanisms has been observed, encompassing an increased expression of ROS-generating enzymes, heightened ROS production, diminished antioxidant bioavailability, elevated peroxynitrite formation, an impaired ADMA–NO pathway, and augmented oxidative damage.

The elevated expression of ROS-generating enzymes and increased ROS production are linked to various CKM syndrome models, including those induced by a maternal high-fructose diet [62], uteroplacental insufficiency [75], maternal stress [82], nicotine exposure [85], and ethanol exposure [90]. Several CKM syndrome models also demonstrate reduced antioxidant bioavailability, such as decreased levels of glutathione [57], and compromised antioxidant enzyme activities [80], including SOD, glutathione peroxidase 1, and catalase.

The formation of 3-nitrotyrosine (3-NT), indicative of peroxynitrite-mediated protein modification [102], is correlated with CKM syndrome in offspring programmed by maternal caloric restriction [54], maternal low-protein diet [58], and prenatal nicotine exposure [87]. Additionally, various biomarkers reflecting oxidative damage to proteins, lipids, and DNA have been assessed in CKM syndrome animal models. These include malondialdehyde (MDA) [62], thiobarbituric-acid-reactive substances (TBARS) [70], F2-isoprostanes [75], 4-hydroxynonenal (4-NHE) [87], and 8-hydroxydeoxyguanosine (8-OHdG) [83,92].

Another contributing mechanism of oxidative stress in the pathogenesis of programmed CKM syndrome is the impaired ADMA–NO pathway. High levels of ADMA, inhibiting NO production and thereby impairing endothelial function, have been associated with the development of various CKM syndrome components. This is evident in models of caloric restriction [53], maternal diabetes [69], maternal stress [79], and perinatal bisphenol A (BPA) exposure [94].

### 2.5. Perinatal Use of Antioxidants as a Reprogramming Strategy

Given the pivotal role of early-life oxidative stress in influencing adverse outcomes in offspring, there has been a notable interest in exploring the perinatal use of antioxidants as a potential preventive strategy for adult diseases [15,24,52,103]. However, conclusive evidence of confirmed benefits from antioxidant supplementation, particularly in pregnant or lactating women, remains elusive in the majority of human studies [104]. The conflicting findings and challenges in interpreting clinical evidence may stem from factors such as the study population, the type of antioxidant, supplement timing and dosage, and the specific disease status deemed suitable for treatment. Hence, it is crucial to conduct additional research aimed at pinpointing the precise developmental stage (such as gestation or lactation) and unraveling organ-specific redox-sensitive signaling pathways associated with various maternal insults that contribute to CKM programming in animal models. This groundwork is essential before translating the findings into clinical applications.

Antioxidants can be classified as natural or synthetic [105]. Examples of natural antioxidants include vitamins A, C, and E, glutathione, polyphenols, carotenoids, and melatonin [106]. Although many natural antioxidants are widely used in humans [104], their benefits in pregnant and lactating women remain inconclusive.

Melatonin, a multifunctional hormone, is implicated in pregnancy and fetal development [107]. Research on melatonin has highlighted its crucial role in antioxidant defense against oxidative damage [26]. Notably, the utilization of melatonin during the perinatal period has been suggested as a reprogramming strategy to mitigate the risk of various adult diseases associated with DOHaD [108]. This summary outlines the current evidence supporting the perinatal use of melatonin to protect against offspring CKM syndrome (Figure 2).

## 3. Melatonin

### 3.1. Effects of Melatonin

Melatonin, initially identified in the bovine pineal gland [109], is predominantly synthesized by pinealocytes using tryptophan as a precursor [110]. While the pineal gland is a primary source, melatonin is also produced in various organs such as the retina, skin, gastrointestinal tract, and bone marrow [25]. The metabolism of melatonin occurs primarily in the liver and kidneys, with its primary urinary metabolite being 6-sulfatoxymelatonin [111].

Melatonin exerts its actions through binding to melatonin receptor-1 (MT1) and -2 (MT2), both of which are G-protein-coupled receptors widely distributed throughout the body [112]. Although there is some controversy regarding retinoid acid receptor (ROR) as a potential nuclear receptor for melatonin [113], the evidence remains inconclusive [114]. Melatonin also exhibits receptor-independent effects, such as activating cytoprotective pathways and functioning as a broad-spectrum antioxidant, possibly through its metabolites [111].

The physiological functions of melatonin are diverse and include the regulation of circadian rhythms, control of blood pressure and autonomic cardiovascular regulation, modulation of the immune system, regulation of energy expenditure and body mass, and vital roles in normal pregnancy and fetal development [25,26,115]. The principal mechanisms underlying melatonin’s antioxidant properties involve scavenging ROS/RNS, promoting the expression of antioxidant enzymes (e.g., SOD and glutathione reductase), and increasing the availability of NO.

### 3.2. Melatonin in Pregnancy and Fetal Development

Melatonin assumes a crucial role in various stages of reproductive processes, encompassing ovulation, fertilization, embryo implantation, and acting as a regulator during pregnancy [116]. In pregnant women, nighttime blood melatonin concentrations are higher than those in their non-pregnant counterparts, peaking at term and returning to physiological levels post-delivery [116]. Maternal melatonin easily traverses the placenta, offering photoperiodic cues to the developing fetus [117]. Beyond the pineal gland, the placenta can independently generate melatonin in a circadian-independent manner [118]. This placental melatonin system plays a pivotal role in scavenging free radicals, thereby attenuating oxidative damage in compromised pregnancies [119]. Experimental evidence supports melatonin’s involvement in ensuring adequate placental perfusion, preventing vascular damage, local inflammation, and oxidative stress [116]. Conversely, a deficiency in maternal melatonin disrupts circadian rhythms, compromises organogenesis, and contributes to IUGR in adult rat offspring [119,120]. Additionally, melatonin exerts biological influence in timing the onset of spontaneous labor and the efficacy of uterine contractions during labor [116]. These findings underscore the indispensable role of melatonin in ensuring normal pregnancy and fetal development. Figure 3 encapsulates the role of melatonin in pregnancy.

### 3.3. Perinatal Melatonin Use in Animal Models of CKM Syndrome of Developmental Origins

Due to its antioxidant effects, melatonin shows potential efficacy in treating disorders associated with CKM syndrome, including obesity [121], diabetes mellitus [121], metabolic syndrome [122], CKD [123], hypertension [124], and CVD [125]. Although certain human studies have investigated the administration of melatonin during pregnancy and lactation, comprehensive reviews highlight the absence of focused assessments on the long-term outcomes for offspring [126,127,128].

To address this gap, Table 2 provides a summary of studies documenting the protective effects of melatonin as a reprogramming strategy in rat models related to CKM syndrome of developmental origins [78,81,129,130,131,132,133,134,135,136,137,138,139]. The therapeutic duration is limited to the perinatal period before the onset of disease. Table 2 includes various early-life insults that lead to offspring CKM syndrome, and perinatal melatonin use has been shown to avert these effects. These insults encompass maternal chronodisruption [77,78], maternal hypertension [129,138], maternal caloric restriction [130], maternal N^G^-nitro-L-arginine-methyl ester (L-NAME) exposure [131], maternal high-methyl-donor diet [132], maternal high-fructose diet [133], maternal high-fat diet [134], perinatal glucocorticoid exposure [81,135,136,137], and maternal hypoxia [139]. Melatonin administration during pregnancy [129,139], lactation [137], or both periods [130,131,132,133,134,135,136] has demonstrated protective actions. Melatonin can be administered through injection [134,137] or via drinking water [78,129,130,131,132,133,134,135,136]. Table 2 showcases the protective effects of perinatal melatonin use in rat offspring aged from 3–27 weeks, corresponding to humans from childhood to young adulthood. In Table 2, the findings from animal studies suggest that melatonin doses within the range of 0.05–10 mg/kg/day demonstrate protective effects for rats.

### 3.4. Effects of Melatonin in Renal Programming

Accumulating evidence indicates that exposure to unfavorable environmental stimuli during kidney development raises the likelihood of CKD and hypertension in adulthood, primarily through renal programming [140,141]. Maternal melatonin deficiency has been implicated in causing renal programming, leading to offspring hypertension [142,143].

In a prior study, we scrutinized the renal transcriptome of male rat offspring born to dams receiving melatonin supplementation throughout pregnancy and lactation, covering the entire period of kidney development [144]. The results revealed 455, 230, and 132 differentially expressed genes in the offspring’s kidneys at 1, 12, and 16 weeks of age, respectively. Maternal melatonin therapy was found to upregulate several epigenetic regulators during kidney development, indicating its potential epigenetic effects [145]. Furthermore, melatonin administration influenced numerous biological pathways associated with kidney development.

Being a widely acknowledged antioxidant, melatonin therapy has exhibited protective effects against oxidative-stress-related renal programming in adult rat offspring. This was observed in models of dams receiving a low-caloric diet [130], antenatal L-NAME administration [131], dams fed with diet rich in methyl donor [132] or fructose [133], and antenatal glucocorticoid administration [135]. Perinatal melatonin use also proved beneficial in terms of renal programming, particularly in rebalancing the ADMA–NO pathway. In a maternal caloric restriction model [130], the protective mechanisms of melatonin against offspring hypertension were associated with a reduction in plasma ADMA, an increase in renal NO levels, and epigenetic changes in numerous genes within the offspring’s kidneys.

### 3.5. Effects of Melatonin in Cardiovascular Programming

While the cardiovascular benefits of antioxidants, in general, await definitive confirmation [136], melatonin has emerged as a potential reprogramming intervention against cardiovascular programming complexities associated with maternal hypoxia [139]. Adult rat offspring exposed to hypoxic pregnancy displayed cardiac wall thinning and heightened vasoconstrictor reactivity, conditions mitigated by maternal treatment with melatonin. The protective effect of melatonin was linked to an increase in the cardiac protein expression of eNOS [139].

It is noteworthy that melatonin’s cardioprotective attributes are also ascribed to its various antioxidant actions [146], including the inhibition of mitochondrial respiratory chain complex [147], the activation of nuclear factor erythroid 2-related factor 2 (Nrf2) [148], and the suppression of inflammatory cytokine release [149]. However, these mechanisms have yet to be thoroughly examined in animal models of oxidative-stress-related cardiovascular programming. A prior study suggested the beneficial role of melatonin in CVD programming in offspring primed by maternal diabetes [150]. While cardiac function and structure were comparable between adult mice offspring born to diabetic and nondiabetic dams, the diabetic offspring exhibited increased infarct size, cardiac dysfunction, and myocardial apoptosis in response to myocardial ischemia/reperfusion, along with heightened oxidative stress. Notably, maternal melatonin supplementation improved myocardial ischemic tolerance in the diabetic offspring.

### 3.6. Effects of Melatonin in Metabolic Programming

As outlined in Table 2, maternal melatonin therapy has demonstrated favorable effects against metabolic-programming-induced insulin resistance [77], obesity [134], hyperglycemia [134], hyperlipidemia [134,137], diabetes [137], and liver steatosis [134,135,137]. In a maternal high-fat diet model, maternal melatonin therapy shielded adult rat offspring from obesity, hyperglycemia, hyperlipidemia, and liver steatosis, concurrently reducing hepatic oxidative stress [134]. Another study indicated that liver steatosis induced by prenatal dexamethasone administration could be averted through maternal melatonin supplementation [135].

Adipose tissues are known to produce pro-inflammatory cytokines and ROS, which play a decisive role in the development of obesity [151]. Melatonin has the potential to counteract the detrimental effects of obesity on perivascular adipose tissue, addressing issues such as excessive ROS production, diminished SOD activity, and reduced NO availability [152]. Also, melatonin could stimulate the Nrf2 signaling pathway to reduce lipopolysaccharide-induced ROS generation [153] and ameliorate H_2_O_2_-induced oxidative stress through the modulation of the extracellular-signal-regulated kinase (ERK)/Akt/nuclear factor κ (NF-κB) pathway [154]. By activating antioxidant enzymes related to glutathione metabolism, melatonin could protect tissues such as the pancreas, adipose tissue, and liver from oxidative stress [151,155], through which it fights obesity-related disorders.

Similarly, in corticosterone-programmed diabetic offspring rats, melatonin demonstrated the ability to prevent metabolic programming [137]. The corticosterone-programmed diabetic offspring exhibited hyperglycemia, hypoinsulinemia, hyperlipidemia, and liver steatosis, accompanied by increased lipid peroxidation and decreased levels of enzymatic antioxidants. However, treatment with melatonin effectively prevented both CKM-syndrome-related components and oxidative stress [137].

### 3.7. Pros and Cons

Although animal studies provide evidence supporting melatonin as a promising reprogramming strategy for preventing CKM syndrome, its efficacy is yet to be validated in human trials. While melatonin is classified as a dietary supplement in the United States, it is only available as a prescription drug in many countries [156].

Currently, oral melatonin supplementation in humans is considered to have a favorable safety profile, with typical doses ranging from 2 to 10 mg in various populations [30]. Experimental data have revealed that melatonin is effective as an antioxidant at high pharmacological doses. Notably, uncertainty surrounds the use, especially at high dosages, of melatonin by mothers during pregnancy and lactation. Clinical studies, thus far, do not recommend the use of melatonin in pregnant and lactating women [30].

In the rat models outlined in Table 2, melatonin administered at a dosage ranging from 0.05 to 10 mg/kg/day to dams demonstrates protective effects against CMK syndrome in their offspring. It is noteworthy that melatonin exhibits remarkable non-toxicity, with a high safety profile observed in rats. The lethal dose 50 for intraperitoneal melatonin injection in rats was determined to be 1168 mg/kg, while the oral administration of melatonin (tested up to 3200 mg/kg) could not be determined [157]. Calculating the human equivalent dose of melatonin for a 75 kg adult based on body surface area normalization from the doses used in various rat models yields a range of 0.6 to 120 mg [158]—approximately one order of magnitude higher than commonly employed doses in humans based on the experiments in Table 2 [159].

Despite these data, there remains a scarcity of information from clinical trials regarding the use and safety of melatonin in pregnant or breastfeeding women [128]. A phase I clinical trial involving 20 pregnant women with early-onset preeclampsia administered a daily total of 30 mg melatonin (in three 10 mg doses) and reported the safety of melatonin for both women and their fetuses, with no adverse events or reactions observed in mothers, fetuses, or neonates [160]. To the best of our knowledge, no trial results focusing on the safety or efficacy of melatonin during pregnancy and lactation specifically for offspring outcomes, especially related to CKM syndrome, have been published [161]. This review underscores the imperative need for well-powered clinical trials in this domain.

As reviewed elsewhere [29], melatonin has been investigated as a potential treatment for various neonatal conditions, encompassing periventricular leukomalacia [162], hypoxic–ischemic injury [163], respiratory distress syndrome [164], and sepsis [165]. Given the challenges of recruiting pregnant women for medical research, utilizing lactating women and their neonates as a reprogramming strategy presents a feasible starting point.

Another concern is that targeted oxidative stress therapy can potentially be harmful. Excessive antioxidant supplementation may shift oxidative stress to the opposite state, known as antioxidant stress [166,167]. Despite its antioxidant property, melatonin may exhibit prooxidant activity as a modulator of cellular redox status [168]. Additionally, healthy tissues not affected by oxidative stress damage may be non-specifically targeted by antioxidants. When melatonin circulates and reaches various organs, it may negatively affect healthy tissues by reducing their levels of ROS below physiologically normal limits. Given the importance of redox homeostasis for normal pregnancy and fetal development, melatonin supplementation as an antioxidant during pregnancy and breastfeeding should only be considered in cases of oxidative stress, not as a routine dietary supplement.

Melatonin operates through various mechanisms to exert its functions. The beneficial effects of perinatal melatonin therapy may be attributed to several mechanisms beyond antioxidants, which are known to interact with oxidative stress underlying CKM syndrome. However, these biological actions may also have negative consequences. In light of this, additional research is essential to determine whether any detrimental consequences emerge from the perinatal administration of melatonin to offspring.

## 4. Concluding Remarks

The ROS generated during fetal and perinatal stages have the potential to influence the long-term health of individuals, increasing their vulnerability to various adult diseases. The impact of oxidative stress extends to key organs such as the kidneys, heart, blood vessels, and metabolic control systems, eventually contributing to CKM syndrome. Despite this understanding, a significant gap remains in animal models that directly explore the consequences of early-life oxidative stress on all aspects of CKM traits in adult offspring. Variations in outcomes may be attributed to diverse environmental insults, the susceptibility of specific developmental windows, organ-specific responses to oxidative stress, or the timing of offspring assessments. Consequently, further research is imperative to comprehend the intricacies of redox-sensitive signaling in different organs, which are affected by various maternal-derived insults during pregnancy and the perinatal period, influencing CKM programming. This knowledge will aid in the development of targeted antioxidant strategies to mitigate the impact of these insults on disease manifestation.

Additionally, melatonin emerges as a potential reprogramming strategy for preventing CKM syndrome. As a well-established antioxidant, melatonin exerts beneficial effects on human health, characterized by a favorable safety profile. Although animal studies showcase the notable advantages of melatonin therapy in CKM programming, its efficacy still awaits validation through human investigations. The anticipation is that melatonin may find application in clinical settings, influencing the interplay between oxidative stress and CKM syndrome in the future.

## Figures and Tables

**Figure 1 antioxidants-13-00226-f001:**
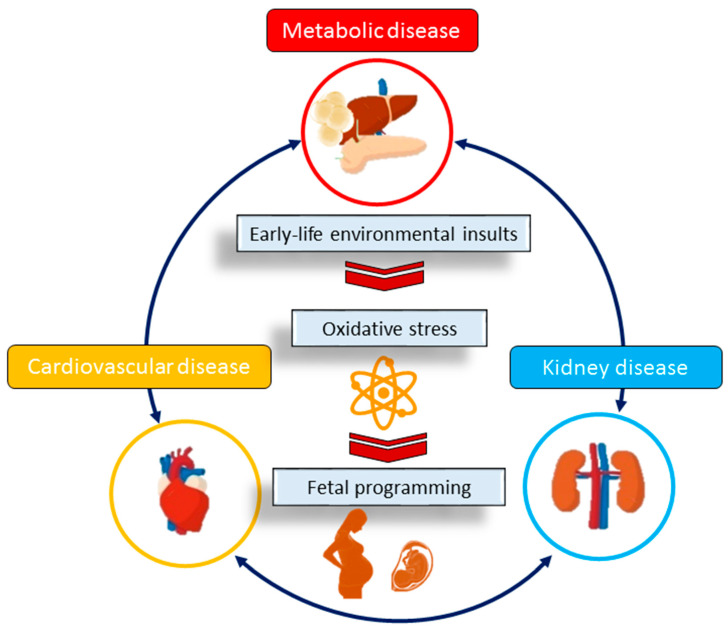
Schematic representation of the interrelationships between early-life oxidative stress, fetal programming, and cardiovascular–kidney–metabolic syndrome in later life.

**Figure 2 antioxidants-13-00226-f002:**
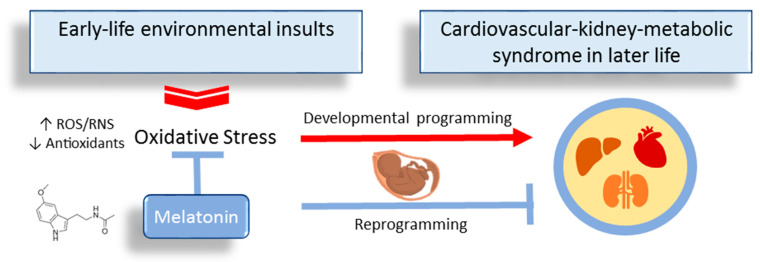
Overview of perinatal use of melatonin as a reprogramming strategy to prevent developmental programming of cardiovascular–kidney–metabolic syndrome.

**Figure 3 antioxidants-13-00226-f003:**
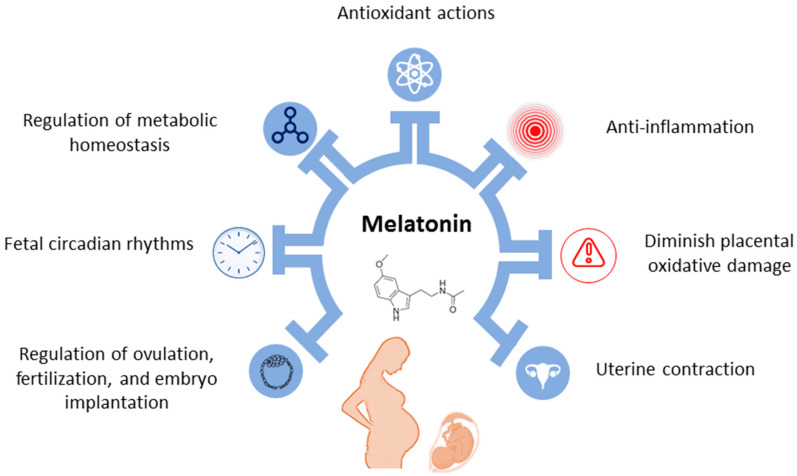
Role of melatonin in pregnancy.

**Table 1 antioxidants-13-00226-t001:** Animal models displaying CKM syndrome in adult offspring related to oxidative stress.

Model	Timing	Age at Evaluation (Weeks)	Features of CKM Syndrome	Oxidative Stress	References
Caloric restriction, 50%	Throughout gestation and lactation	12–16	Hypertension, insulin resistance, and kidney disease	↑Renal 8-OHdG expression, ↑3-NT, ↑ADMA, ↓NO	[53,54,55]
Protein restriction, 6–9%	Gestation	12	Hypertension, insulin resistance, and kidney disease	↑F_2_-isoprostane, ↑3-NT, ↓SOD and GPX activity, ↓glutathione	[56,57,58]
Maternal high-fructose diet, 60%	Throughout gestation and lactation	12–52	Hypertension, obesity, insulin resistance, and dyslipidemia	↑Renal 8-OHdG expression ↑MDA, ↑Brain NADPH-oxidase expression; ↑ROS, ↓NO	[59,60,61,62,63]
Maternal high-fat diet, 58%	Throughout gestation and lactation	16	Hypertension, obesity, insulin resistance, dyslipidemia, and kidney disease	↓SOD activity, ↑Renal MDA; ↑Renal 8-OHdG expression	[64,65,66,67,68]
Maternal diabetes	Neonatal streptozotocin injection	12–16	Hypertension, obesity, insulin resistance, dyslipidemia, and kidney disease	↑Renal 3-NT and TBARS; ↑ROS, ↓SOD activity, ↓NO; ↑ADMA	[69,70,71]
Uteroplacental insufficiency	Bilateral uterine artery ligation on gestation	22–30	Hypertension, dyslipidemia insulin resistance, and kidney disease	↑Renal NADPH-oxidase dependent superoxide, ↑Urinary F_2_-isoprostane level	[72,73,74,75]
Maternal chronodisruption	Gestation	12–52	Hypertension and insulin resistance	↑Brain ROS	[76,77,78]
Maternal stress	Perinatal dexamethasone administration	16–24	Hypertension, obesity, insulin resistance, and kidney disease	↓Gpx1 expression, ↑NADPH-oxidase, ↓Renal NO, ↑ADMA, ↑Renal 8-OHdG expression	[79,80,81,82]
Maternal chronic kidney disease	Throughout gestation and lactation	12	Hypertension and kidney disease	↑Renal 8-OHdG expression, ↓NO	[83,84]
Nicotine exposure	Perinatal nicotine exposure	20–32	Hypertension, hyperlipidemia, steatosis, and kidney disease	↑MDA, ↑3-NT, ↑NADPH oxidase, ↑4-NHE, ↓GPx1 activity	[85,86,87,88]
Ethanol exposure	Gestation	24	Hypertension and insulin resistance	↓SOD1, ↓CAT, ↓GPX, ↑NOX2	[89,90,91]
TCDD exposure	Gestation and lactation	12	Hypertension, cardiac hypertrophy, and kidney disease	↑Renal 8-OHdG expression	[92,93]
BPA exposure	Gestation and lactation	16–24	Hypertension, insulin resistance, and steatosis	↓SOD, ↓CAT, ↑Renal 8-OHdG expression, ↑ADMA, ↓NO	[94,95,96]
DEHP exposure	Gestation and lactation	12–21	Hypertension and insulin resistance	↑Renal ROS, ↑Renal 8-OHdG expression	[97,98,99,100]

TCDD = 2,3,7,8-tetrachlorodibenzo-p-dioxin; BPA = bisphenol A; DEHP = di-n-butyl phthalate; ADMA = asymmetric dimethylarginine; 8-OHdG = 8-hydroxy-2’-deoxyguanosine; NO = nitric oxide; ROS = reactive oxygen species; 3-NT = 3-nitrotyrosine; TBARS = thiobarbituric acid; MDA = malondialdehyde; 4-NHE = 4-hydroxynonenal; CAT = catalase; GPX = glutathione peroxidase; SOD = superoxidase dismutase; NOX2 = NADPH oxidase 2.

**Table 2 antioxidants-13-00226-t002:** Protective effects of perinatal melatonin supplementation against CKM syndrome in rat models.

Treatment Period and Dose	Model	Age at Evaluation (Weeks)	Prevented CKM Syndrome in Offspring	Ref.
0.5 mg/kg/day p.o. throughout gestation and lactation	Maternal chronodisruption	18	Insulin resistance	[77]
10 mg/kg/day p.o. throughout gestation and lactation	Maternal chronodisruption	12	Hypertension	[78]
10 mg/kg/day p.o. during pregnancy	Maternal hypertension	8	Hypertension	[129]
10 mg/kg/day p.o. throughout gestation and lactation	Caloric restriction	12	Hypertension and kidney disease	[130]
10 mg/kg/day p.o. throughout gestation and lactation	Maternal L-NAME exposure	12	Hypertension and kidney disease	[131]
10 mg/kg/day p.o. throughout gestation and lactation	Maternal high methyl-donor diet	12	Hypertension	[132]
10 mg/kg/day p.o. throughout gestation and lactation	Maternal high-fructose diet	12	Hypertension	[133]
5 mg/kg/day i.p. throughout gestation and lactation	Maternal high-fat diet	3	Obesity, hyperglycemia, hyperlipidemia, and liver steatosis	[134]
10 mg/kg/day p.o. throughout gestation and lactation	Prenatal dexamethasone exposure	16	Hypertension, liver steatosis, and kidney disease	[81,135]
10 mg/kg/day p.o. throughout gestation and lactation	Neonatal dexamethasone exposure	16	Hypertension	[136]
1 mg/kg/day at night from postnatal day 2 to 14	Neonatal corticosterone exposure	16	Diabetes, hyperlipidemia, liver steatosis, and kidney disease	[137]
0.2 mg/kg/day p.o. throughout gestation and lactation	Maternal hypertension	27	Hypertension	[138]
0.05 mg/kg/day p.o.during gestation	Maternal hypoxia	16	Cardiovascular disease	[139]

p.o. = per oral; i.p. = intraperitoneal; L-NAME = N^G^-nitro-l-arginine methyl ester.

## Data Availability

Data are contained within the article.

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
