# Peer review of "Melatonin Use during Pregnancy and Lactation Complicated by Oxidative Stress: Focus on Offspring’s Cardiovascular–Kidney–Metabolic Health in Animal Models"

_antioxidants, 2024, doi:10.3390/antiox13020226_

Round 1

Reviewer 1 Report

Comments and Suggestions for Authors

This review first provides an overview of oxidative stress mechanisms that throughout pregnancy can affect developmental processes which in turn can lead to developmental programming of adult diseases, including CKM syndrome. Then, the authors report current knowledge about the melatonin roles in pregnancy and fetal development suggesting its perinatal use as reprogramming Strategy to prevent adult disease. Furthermore, the authors make a short dissertation on pros and cons of melatonin application based on current data.

The topic is interesting, the manuscript is well-structured and well-written with a fluent language easy to understand, and the sections briefly but clearly present available literature data.

In my opinion this manuscript is suitable for publication.

Author Response

RESPONSES TO REVIEWER’S COMMENTS

Reviewer #1

This review first provides an overview of oxidative stress mechanisms that throughout pregnancy can affect developmental processes which in turn can lead to developmental programming of adult diseases, including CKM syndrome. Then, the authors report current knowledge about the melatonin roles in pregnancy and fetal development suggesting its perinatal use as reprogramming Strategy to prevent adult disease. Furthermore, the authors make a short dissertation on pros and cons of melatonin application based on current data.

The topic is interesting, the manuscript is well-structured and well-written with a fluent language easy to understand, and the sections briefly but clearly present available literature data.

In my opinion this manuscript is suitable for publication.

RESPONSE: We thank Reviewer #1 for his/her generous support.

Reviewer 2 Report

Comments and Suggestions for Authors

This is an interesting and well-written review about the effects of melatonin during the perinatal period. The review focuses mostly on the so-called Cardiovascular-kidney-metabolic (CKM) syndrome (also known as cardiometabolic renal syndrome (CMR)) a new term recently coined by the American Heart Association (AHA) which involves the co-occurrence of chronic diseases of the kidney (CKD), metabolic risk factors such as obesity and diabetes type 2, and cardiovascular disease. CKM is a very common syndromic progressive condition beginning in early life with related signs and symptoms, implying a shared underlying pathophysiology. CKM syndrome has the potential to negatively affect any organ in the body, and it is as such difficult to define properly. I think that this review would benefit from a better discussion of what exactly is CMK, namely not specifically Cardiovascular, Kidney and/or metabolic, but the relation between these different conditions. The focus on the connection between these three syndromic aspects is essential,  considering them separately has been the object of many other reviews also from the same authors which are not necessarily cited (e.g. Int. J. Mol. Sci. 2019, 20, 5681; doi:10.3390/ijms20225681). Otherwise I find the review interesting and timely, but as it stands it misses the possibility to focus on the important health issue that CMK constitutes and, in my view, can be improved.

Author Response

Reviewer #2

This is an interesting and well-written review about the effects of melatonin during the perinatal period. The review focuses mostly on the so-called Cardiovascular-kidney-metabolic (CKM) syndrome (also known as cardiometabolic renal syndrome (CMR)) a new term recently coined by the American Heart Association (AHA) which involves the co-occurrence of chronic diseases of the kidney (CKD), metabolic risk factors such as obesity and diabetes type 2, and cardiovascular disease. CKM is a very common syndromic progressive condition beginning in early life with related signs and symptoms, implying a shared underlying pathophysiology. CKM syndrome has the potential to negatively affect any organ in the body, and it is as such difficult to define properly.

RESPONSE: We thank the reviewer #2 for the efforts and the constructive comments on the work.

I think that this review would benefit from a better discussion of what exactly is CMK, namely not specifically Cardiovascular, Kidney and/or metabolic, but the relation between these different conditions. The focus on the connection between these three syndromic aspects is essential, considering them separately has been the object of many other reviews also from the same authors which are not necessarily cited (e.g. Int. J. Mol. Sci. 2019, 20, 5681; doi:10.3390/ijms20225681).

RESPONSE: In response to the recommendation, we have revised our statements to more effectively address the interconnections within CKM and have omitted the specific reference you mentioned.

P1, lines 31-56:

“Increasing recognition of the pathological interconnections among metabolic risk factors like obesity and diabetes, chronic kidney disease (CKD), and cardiovascular dis-ease (CVD) has given rise to the conceptualization of cardiovascular-kidney-metabolic (CKM) syndrome [1]. For the first time, the 2023 Scientific Statement from the American Heart Association defines CKM syndrome as a systemic disorder marked by pathophysiological interactions among metabolic risk factors, CKD, and the cardiovascular system. This interaction results in multiorgan dysfunction and a heightened risk of adverse cardiovascular and renal outcomes [1]. CKM syndrome is classified into four distinct stages, ranging from stage 0 to stage 4. These stages likely represent varying degrees of progression and severity within the complex spectrum of this disease. Various key components manifest at different stages, contributing to the varied degrees of progression and severity within the complex spectrum of CKM syndrome.

While the global disease burden of individual metabolic diseases, CKD, and CVD is widely recognized [2-5], the complex and mutually reinforcing detrimental relationships among them remain largely unclear. Epidemiological studies suggest a multidirectional connection among these conditions. For instance, individuals with CKD face an elevated risk of developing CVD [6]. Comorbidities associated with CKD, such as obesity, diabetes, and hypertension, constitute significant components of metabolic syndrome. [7]. Additionally, cardiorenal syndrome is a recognized condition in which dysfunction of either the heart or kidneys can impair function in the other organ [8]. Furthermore, metabolic syndrome, a cluster of conditions, collectively amplifies the susceptibility to CVD [9]. Recently, the concept of metabolic syndrome has evolved into the cardiometabolic syndrome, encompassing characteristics of metabolic syndrome along with traditional CVD factors [10]. Consequently, it is plausible that CVD, CKD, and metabolic diseases may interact at the pathophysiological level, leading to clinical overlap between these health conditions [11].”

Otherwise I find the review interesting and timely, but as it stands it misses the possibility to focus on the important health issue that CMK constitutes and, in my view, can be improved.

RESPONSE: Our statements have been rephrased to underscore this matter.

P2, lines 60-66:

“Although managing the entire syndrome holistically, rather than focusing on individual diseases, is recommended for optimal care [12], a comprehensive therapeutic guideline is still pending. It is noteworthy that placing emphasis on early prevention has the potential to alleviate the burden associated with CKM syndrome. Recognizing the interplay be-tween CKM diseases is crucial for adopting a more holistic approach to CKM care, moving beyond isolated treatment of individual conditions. This broader perspective holds promise for enhancing global health outcomes in the future.”

Reviewer 3 Report

Comments and Suggestions for Authors

Sorry for the slight delay in my assessment, but I had to read this paper several times to try to better capture the paper as submitted. Overall, this rather reads as a narrative review on the topic, and this might be a priority issue for this paper, and further builds on previous reviews on the topic published by the same authors (Antioxidants 2022a and 2022b; Int J Mol Sc 2019; Int J Mol Sc 2017). It is not clear to me how ‘new’ this current paper is to the previous papers of the same authors. Related to this, there is a lot of referral to own previous references.

Only after reading the full paper, it was clear that the focus is on preclinical animal models, and this should be much clearer in the title and abstract. Along the same line, there is no reflection on a ‘dose’ effect approach (not quantitative).

It is not clear how the search strategy has been developed and has been done to construct the tables ?

Author Response

Reviewer #3

Sorry for the slight delay in my assessment, but I had to read this paper several times to try to better capture the paper as submitted. Overall, this rather reads as a narrative review on the topic, and this might be a priority issue for this paper, and further builds on previous reviews on the topic published by the same authors (Antioxidants 2022a and 2022b; Int J Mol Sc 2019; Int J Mol Sc 2017). It is not clear to me how ‘new’ this current paper is to the previous papers of the same authors. Related to this, there is a lot of referral to own previous references.

RESPONSE: We are grateful to the Reviewer #3 for your careful review and valuable feedback on our manuscript. Approximately 1500 reviews focus on oxidative stress in pregnancy, with fewer than 200 specifically addressing offspring outcomes. While we have previously explored the impact of oxidative stress in DOHaD-related diseases, such as hypertension or kidney disease, this marks our inaugural emphasis on offspring's CKM syndrome.

The novelty of this review lies in being the first and only one to examine the connections among melatonin, maternal oxidative stress, and the development of offspring CKM syndrome. The official definition of this syndrome by the AHA in 2023 underscores the timeliness and urgency of this review, contributing to its considerable scientific interest. As pioneers in this emerging field, we find it necessary to cite our own original studies due to a scarcity of alternative sources, and we genuinely hope the reviewer comprehends our situation. Nevertheless, we kindly request knowledgeable reviewers to offer additional references related to this topic in case we have overlooked any.

Only after reading the full paper, it was clear that the focus is on preclinical animal models, and this should be much clearer in the title and abstract.

RESPONSE: Currently, there is a lack of information regarding oxidative stress during human pregnancy that may be associated with offspring CKM syndrome in childhood or adulthood. Consequently, our data primarily originates from animal studies, a point we have highlighted in the abstract and title to address this limitation.

P1, lines 24-26:

“Currently, there is a lack of sufficient information in humans, and the available evidence primarily originates from animal studies.”

Along the same line, there is no reflection on a ‘dose’ effect approach (not quantitative).

RESPONSE: Following the recommendation, we have incorporated the following statements to elaborate on the melatonin dosage.

P11, lines 391-409:

“In the rat models outlined in Table 2, melatonin administered at a dosage ranging from 0.05 to 10 mg/kg/day to dams demonstrates protective effects against CMK syndrome in their offspring. It is noteworthy that melatonin exhibits remarkable non-toxicity, with a high safety profile observed in rats. The lethal dose 50 for intraperitoneal melatonin injection in rats was determined to be 1168 mg/kg, while the oral administration of melatonin (tested up to 3200 mg/kg) could not be determined [157]. Calculating the human equivalent dose of melatonin for a 75 kg adult based on body surface area normalization from the doses used in various rat models yields a range of 0.6 to 120 mg [158]—approximately one order of magnitude higher than commonly employed doses in humans based on the experiments in Table 2 [159].

Despite this data, there remains a scarcity of information from clinical trials regard-ing the use and safety of melatonin in pregnant or breastfeeding women [128]. A phase I clinical trial involving 20 pregnant women with early-onset preeclampsia, administered a daily total of 30 mg melatonin (in three 10 mg doses), reported the safety of melatonin for both women and their fetuses, with no adverse events or reactions observed in mothers, fetuses, or neonates [160]. To the best of our knowledge, no trial results focusing on the safety or efficacy of melatonin during pregnancy and lactation specifically for offspring outcomes, especially related to CKM syndrome, have been published [161]. This review underscores the imperative need for well-powered clinical trials in this domain.”

It is not clear how the search strategy has been developed and has been done to construct the tables ?

RESPONSE: We included the following paragraph to outline our literature search strategy, incorporating keywords relevant to CKM syndrome.

P3, lines 94-105:

“A comprehensive literature review was conducted by querying the databases Embase, MEDLINE, and Cochrane Library to identify studies published in English. The search utilized relevant keywords pertaining to melatonin, oxidative stress, DOHaD, and CKM syndrome. The employed search terms included “metabolic syndrome”, “hypertension”, “dyslipidemia”, “hyperlipidemia”, “obesity”, “diabetes”, “insulin resistance”, “hyperglycemia”, “liver steatosis”, “chronic kidney disease”, “cardiovascular disease”, “atherosclerosis”, “heart failure”, “cardiorenal syndrome”, “developmental programming”, “DOHaD”,“ free radicals”, “offspring”, “progeny”, “mother”, “prenatal”, “nitric oxide”, “oxidative stress”, “pregnancy”, “reprogramming”, “reactive oxygen species”, “reactive nitrogen species”, and “melatonin”. Additionally, we scrutinized the reference lists of articles to identify any additional references relevant to this review. The search concluded on November 30, 2023.”

Reviewer 4 Report

Comments and Suggestions for Authors

Melatonin Use during Pregnancy and Lactation Complicated 2 by Oxidative Stress: Focus on Offspring’s Cardiovascular-Kidney-Metabolic Health

By You-Lin Tain and Chien-Ning Hsu

This is a well written review, encompassing current knowledge on the development of CKM already during fetal development, and possible beneficial effects of melatonin to mitigate these problems.

Major comments:

Line 43-44  ‘leading to chronic diseases later in life’.  This is an overstatement. Not every ‘environmental challenge’ will lead to a disease, let alone a chronic disease later in life. Please adapt this statement.

Line 47 The word ‘excessive’ should be removed here, the word ‘imbalance’ suffices.

Line 51 ‘Refer to Figure 1 for a detailed schematic.’ Seems an internal remark and should be removed.

Table 2: the levels of melatonin used for treatment are in different units in this table: as percentage, as mg/ml, as mg/kg/day in drinking water, as mg/kg/day i.p.  In order to compare the different studies, the units should be converted to one (either percentage, or mg/ml, at least for the studies where melatonin was present in the drinking water.

Different for types/sizes are used in the list of references. For instance refs28, 29, 30 are of different size. Please correct.

Author Response

Reviewer #4

Melatonin Use during Pregnancy and Lactation Complicated 2 by Oxidative Stress: Focus on Offspring’s Cardiovascular-Kidney-Metabolic Health By You-Lin Tain and Chien-Ning Hsu

This is a well written review, encompassing current knowledge on the development of CKM already during fetal development, and possible beneficial effects of melatonin to mitigate these problems.

RESPONSE: We express our gratitude to Reviewer #4 for his/her kind assistance.

Major comments:

Line 43-44  ‘leading to chronic diseases later in life’.  This is an overstatement. Not every ‘environmental challenge’ will lead to a disease, let alone a chronic disease later in life. Please adapt this statement.

RESPONSE: We have tempered our statement as follows.

P2, lines 69-72:

“Termed the Developmental Origins of Health and Disease (DOHaD), this theory posits that a developing fetus responds to environmental challenges by making structural and functional adaptations, thereby increasing the risk of developing chronic diseases later in life [14].”

Line 47 The word ‘excessive’ should be removed here, the word ‘imbalance’ suffices.

RESPONSE: Correction has been made.

Line 51 ‘Refer to Figure 1 for a detailed schematic.’ Seems an internal remark and should be removed.

RESPONSE: We have removed this sentence.

Table 2: the levels of melatonin used for treatment are in different units in this table: as percentage, as mg/ml, as mg/kg/day in drinking water, as mg/kg/day i.p.  In order to compare the different studies, the units should be converted to one (either percentage, or mg/ml, at least for the studies where melatonin was present in the drinking water.

RESPONSE: We have transformed the melatonin doses into mg/kg/day for each study presented in Table 2. Additionally, we have incorporated the following information into the text:

P9, lines 301-303:

" In Table 2, the findings from animal studies suggest that melatonin doses within the range of 0.05-10 mg/kg/day demonstrate protective effects for rats."

Different for types/sizes are used in the list of references. For instance refs28, 29, 30 are of different size. Please correct.

RESPONSE: Apologies for the typos. We have corrected our reference format.

Reviewer 5 Report

Comments and Suggestions for Authors

It focuses on the study of cardiovascular-kidney-metabolic (CKM) syndrome and how exposure to oxidative stress in the early stages of life can condition the programming of diseases in adulthood, including the aforementioned syndrome. The use of antioxidants can prevent this programming effect and among these antioxidants is melatonin. The authors carry out a review on the role of oxidative stress during pregnancy and lactation on CKM and how the use of melatonin, due to its antioxidant effect, could be a promising reprogramming tool.

The main limitation that I observe in this publication is the syndrome studied itself. As indicated in a review cited by the authors from 2023 (quote 1 of the article), the definition of this syndrome needs to be clarified, that is, there are multiple definitions based on different expert opinions and this makes it difficult to establish the collection of signs or symptoms. related to this syndrome. If it is not defined correctly we cannot see exactly if, as the title of the work indicates, the preventive use of melatonin is effective or not for this specific syndrome, yes for other syndromes or pathologies, but not for this one.

I think that the first thing that should be done is to establish clearer criteria on the set of signs or symptoms that are going to be considered to establish the CKM—oxidative stress-melatonin relationship.

​Thus, the authors describe oxidative stress, its importance in pregnancy and its relationship with multiple maternal-fetal pathologies and even its role in the appearance of subsequent pathologies in adulthood. They also review that the use of antioxidants, including melatonin, can have a preventive nature on these effects. But this information does not present scientific interest, given that it is something that has already been reviewed multiple times, even by the same authors, who provide 32 citations of their own in this review.

The contribution of greatest scientific interest would be found in the effect or role of oxidative stress in this syndrome and in the fact that melatonin could be used from a preventive point of view, but since the definition and set of signs or symptoms to consider are not clear, that objective does not seem to have weight.

In short, this review seems to be reduced to studying the role of oxidative stress in pregnancy and lactation in multiple disorders, pathologies or syndromes that the authors consider as factors of CKM, but not in themselves that they are CKM. For example, they talk about models of metabolic syndrome, insulin resistance, hypertension, but they are studies or rat models focused on a specific factor and not on the set of factors that can be considered for this syndrome. The authors on page 5, lines 152-154 indicate: “Elevated expression of ROS-generating enzymes and increased ROS production are 151 linked to various CKM syndrome models, including those induced by a maternal high-fructose diet [52], uteroplacental insufficiency [65], maternal stress [72], nicotine exposure 153 [75], and ethanol exposure [80].", but these publications are not rat models for this syndrome, as the authors indicate, but for other pathologies or syndromes already studied.

I think that the scientific contribution, except for the title, is scarce, unless they clarify the definition of this syndrome.

Author Response

Reviewer #5

It focuses on the study of cardiovascular-kidney-metabolic (CKM) syndrome and how exposure to oxidative stress in the early stages of life can condition the programming of diseases in adulthood, including the aforementioned syndrome. The use of antioxidants can prevent this programming effect and among these antioxidants is melatonin. The authors carry out a review on the role of oxidative stress during pregnancy and lactation on CKM and how the use of melatonin, due to its antioxidant effect, could be a promising reprogramming tool.

RESPONSE: We appreciate Reviewer #5 for conducting a thorough review and providing valuable feedback on our manuscript.

The main limitation that I observe in this publication is the syndrome studied itself. As indicated in a review cited by the authors from 2023 (quote 1 of the article), the definition of this syndrome needs to be clarified, that is, there are multiple definitions based on different expert opinions and this makes it difficult to establish the collection of signs or symptoms. related to this syndrome. If it is not defined correctly we cannot see exactly if, as the title of the work indicates, the preventive use of melatonin is effective or not for this specific syndrome, yes for other syndromes or pathologies, but not for this one.

RESPONSE: To enhance clarity regarding this new syndrome, we have incorporated the following statements.

P1, lines 31-56:

“Increasing recognition of the pathological interconnections among metabolic risk factors like obesity and diabetes, chronic kidney disease (CKD), and cardiovascular dis-ease (CVD) has given rise to the conceptualization of cardiovascular-kidney-metabolic (CKM) syndrome [1]. For the first time, the 2023 Scientific Statement from the American Heart Association defines CKM syndrome as a systemic disorder marked by pathophysiological interactions among metabolic risk factors, CKD, and the cardiovascular system. This interaction results in multiorgan dysfunction and a heightened risk of adverse cardiovascular and renal outcomes [1]. CKM syndrome is classified into four distinct stages, ranging from stage 0 to stage 4. These stages likely represent varying degrees of progression and severity within the complex spectrum of this disease. Various key components manifest at different stages, contributing to the varied degrees of progression and severity within the complex spectrum of CKM syndrome.

While the global disease burden of individual metabolic diseases, CKD, and CVD is widely recognized [2-5], the complex and mutually reinforcing detrimental relationships among them remain largely unclear. Epidemiological studies suggest a multidirectional connection among these conditions. For instance, individuals with CKD face an elevated risk of developing CVD [6]. Comorbidities associated with CKD, such as obesity, diabetes, and hypertension, constitute significant components of metabolic syndrome. [7]. Additionally, cardiorenal syndrome is a recognized condition in which dysfunction of either the heart or kidneys can impair function in the other organ [8]. Furthermore, metabolic syndrome, a cluster of conditions, collectively amplifies the susceptibility to CVD [9]. Recently, the concept of metabolic syndrome has evolved into the cardiometabolic syndrome, encompassing characteristics of metabolic syndrome along with traditional CVD factors [10]. Consequently, it is plausible that CVD, CKD, and metabolic diseases may interact at the pathophysiological level, leading to clinical overlap between these health conditions [11].”

I think that the first thing that should be done is to establish clearer criteria on the set of signs or symptoms that are going to be considered to establish the CKM—oxidative stress-melatonin relationship.

RESPONSE: We included the following paragraph to outline our literature search strategy, incorporating keywords relevant to CKM syndrome

P3, lines 94-105:

“A comprehensive literature review was conducted by querying the databases Embase, MEDLINE, and Cochrane Library to identify studies published in English. The search utilized relevant keywords pertaining to melatonin, oxidative stress, DOHaD, and CKM syndrome. The employed search terms included “metabolic syndrome”, “hypertension”, “dyslipidemia”, “hyperlipidemia”, “obesity”, “diabetes”, “insulin resistance”, “hyperglycemia”, “liver steatosis”, “chronic kidney disease”, “cardiovascular disease”, “atherosclerosis”, “heart failure”, “cardiorenal syndrome”, “developmental programming”, “DOHaD”,“ free radicals”, “offspring”, “progeny”, “mother”, “prenatal”, “nitric oxide”, “oxidative stress”, “pregnancy”, “reprogramming”, “reactive oxygen species”, “reactive nitrogen species”, and “melatonin”. Additionally, we scrutinized the reference lists of articles to identify any additional references relevant to this review. The search concluded on November 30, 2023.”

​Thus, the authors describe oxidative stress, its importance in pregnancy and its relationship with multiple maternal-fetal pathologies and even its role in the appearance of subsequent pathologies in adulthood. They also review that the use of antioxidants, including melatonin, can have a preventive nature on these effects. But this information does not present scientific interest, given that it is something that has already been reviewed multiple times, even by the same authors, who provide 32 citations of their own in this review.

RESPONSE: There are approximately 1500 reviews addressing oxidative stress in pregnancy, but only fewer than 200 reviews are specifically related to offspring outcomes. Despite having previously examined the impact of oxidative stress in DOHaD-related diseases, this is our inaugural focus on offspring's CKM syndrome. The official definition of this syndrome by the AHA in 2023 emphasizes the timeliness and urgency of this review, contributing to its significant scientific interest. Being pioneers in this emerging field, we have cited our studies due to a scarcity of alternative sources. We genuinely hope the reviewer comprehends our situation. However, we kindly request knowledgeable reviewers to provide additional references related to this topic in case we have overlooked any. 

The contribution of greatest scientific interest would be found in the effect or role of oxidative stress in this syndrome and in the fact that melatonin could be used from a preventive point of view, but since the definition and set of signs or symptoms to consider are not clear, that objective does not seem to have weight.

RESPONSE: As per the suggestion, we have included the definition of CKM syndrome, the search criteria used, and our unique scientific contribution to enhance comprehension.

​In short, this review seems to be reduced to studying the role of oxidative stress in pregnancy and lactation in multiple disorders, pathologies or syndromes that the authors consider as factors of CKM, but not in themselves that they are CKM. For example, they talk about models of metabolic syndrome, insulin resistance, hypertension, but they are studies or rat models focused on a specific factor and not on the set of factors that can be considered for this syndrome. The authors on page 5, lines 152-154 indicate: “Elevated expression of ROS-generating enzymes and increased ROS production are 151 linked to various CKM syndrome models, including those induced by a maternal high-fructose diet [52], uteroplacental insufficiency [65], maternal stress [72], nicotine exposure 153 [75], and ethanol exposure [80].", but these publications are not rat models for this syndrome, as the authors indicate, but for other pathologies or syndromes already studied.

RESPONSE: CKM syndrome is a complex spectrum of systemic disease, where various key components manifest at different stages. Therefore, the studies listed in Table 1 primarily aggregate findings from animal models displaying at least two components of CKM syndrome in offspring, which are linked to oxidative stress. It's essential to note that the cited publications do not name rat models as CKM syndrome, as this syndrome has just been officially defined since 2023. Nevertheless, all the referenced studies are pertinent to the key components of CKM syndrome in offspring.

I think that the scientific contribution, except for the title, is scarce, unless they clarify the definition of this syndrome.

RESPONSE: By furnishing details to elucidate the definition of CKM syndrome, the search criteria employed, and the distinctive scientific contribution, we earnestly anticipate that this review meets the reviewer's requirements and secures approval.

Round 2

Reviewer 2 Report

Tje authors have now improved their paper particularly regarding the description of the integration of the signs of CMK syndrome.

All suggested changes have been made

Author Response

Reviewer #2

All suggested changes have been made.

RESPONSE: We would like to express our gratitude for the valuable assistance provided by the reviewer.

Reviewer 3 Report

Comments and Suggestions for Authors

The priority and high auto-citation are editorial office aspects to consider. The authors have provided the revisions in line with my questions. Nothing to add. 

Author Response

Reviewer #3

The priority and high auto-citation are editorial office aspects to consider. The authors have provided the revisions in line with my questions. Nothing to add.

RESPONSE: In response to the feedback provided, we have addressed the concerns raised by Reviewer 3 and reduced the self-citation ratio to 14.9%.

Reviewer 4 Report

All questions regarding the previous version of the manuscript have been satisfactorily answered, and the manuscript has been adequately adapted.

No comments

Author Response

Reviewer #4

All questions regarding the previous version of the manuscript have been satisfactorily answered, and the manuscript has been adequately adapted.

RESPONSE: We want to convey our appreciation for the invaluable assistance provided by the reviewer.

Reviewer 5 Report

Comments and Suggestions for Authors

of greatest scientific interest would be found in the effect or role of oxidative stress in this syndrome and in the fact that melatonin could be used from a preventive point of view, but since the definition and set of signs or symptoms to consider are not clear, that objective does not seem to have weight.

In short, this review seems to be reduced to studying the role of oxidative stress in pregnancy and lactation in multiple disorders, pathologies or syndromes that the authors consider as factors of CKM, but not in themselves that they are CKM. For example, they talk about models of metabolic syndrome, insulin resistance, hypertension, but they are studies or rat models focused on a specific factor and not on the set of factors that can be considered for this syndrome.

In short, I still consider that the only novelty that this article provides is adding this pathology to the title, although I do not see anything new in the content of the article.

Author Response

Reviewer #5

of greatest scientific interest would be found in the effect or role of oxidative stress in this syndrome and in the fact that melatonin could be used from a preventive point of view, but since the definition and set of signs or symptoms to consider are not clear, that objective does not seem to have weight.

In short, this review seems to be reduced to studying the role of oxidative stress in pregnancy and lactation in multiple disorders, pathologies or syndromes that the authors consider as factors of CKM, but not in themselves that they are CKM. For example, they talk about models of metabolic syndrome, insulin resistance, hypertension, but they are studies or rat models focused on a specific factor and not on the set of factors that can be considered for this syndrome.

In short, I still consider that the only novelty that this article provides is adding this pathology to the title, although I do not see anything new in the content of the article.

RESPONSE: We remain grateful to Reviewer #5 for offering valuable feedback on our manuscript. With the support of the positive evaluations from the other four reviewers, we are optimistic that this review aligns with the reviewer's criteria and ultimately garners approval.
